# Homo Sapiens (Hsa)-microRNA (miR)-6727-5p Contributes to the Impact of High-Density Lipoproteins on Fibroblast Wound Healing In Vitro

**DOI:** 10.3390/membranes12020154

**Published:** 2022-01-27

**Authors:** Khaled Mahmoud Bastaki, Jamie Maurice Roy Tarlton, Richard James Lightbody, Annette Graham, Patricia Esther Martin

**Affiliations:** Department of Biological and Biomedical Sciences, School of Health and Life Sciences, Glasgow Caledonian University, Glasgow G4 0BA, UK; khaled.bastaki@gcu.ac.uk (K.M.B.); Jamie.tarlton@gcu.ac.uk (J.M.R.T.); Richard.Lightbody@gcu.ac.uk (R.J.L.); Patricia.Martin@gcu.ac.uk (P.E.M.)

**Keywords:** high-density lipoproteins, adult fibroblast, neonatal fibroblast, wound healing, microRNA

## Abstract

Chronic, non-healing wounds are a significant cause of global morbidity and mortality, and strategies to improve delayed wound closure represent an unmet clinical need. High-density lipoproteins (HDL) can enhance wound healing, but exploitation of this finding is challenging due to the complexity and instability of these heterogeneous lipoproteins. The responsiveness of primary human neonatal keratinocytes, and neonatal and human dermal fibroblasts (HDF) to HDL was confirmed by cholesterol efflux, but promotion of ‘scrape’ wound healing occurred only in primary human neonatal (HDFn) and adult fibroblasts (HDFa). Treatment of human fibroblasts with HDL induced multiple changes in the expression of small non-coding microRNA sequences, determined by microchip array, including hsa-miR-6727-5p. Intriguingly, levels of hsa-miR-6727-5p increased in HDFn, but decreased in HDFa, after exposure to HDL. Delivery of a hsa-miR-6727-5p mimic elicited repression of different target genes in HDFn (*ZNF584*) and HDFa (*EDEM3*, *KRAS*), and promoted wound closure in HDFn. By contrast, a hsa-miR-6727-5p inhibitor promoted wound closure in HDFa. We conclude that HDL treatment exerts distinct effects on the expression of hsa-miR-6727-5p in neonatal and adult fibroblasts, and that this is a sequence which plays differential roles in wound healing in these cell types, but cannot replicate the myriad effects of HDL.

## 1. Introduction

Chronic, non-healing wounds are a significant cause of morbidity and mortality in the United Kingdom; according to The Health Improvement Network (THIN database), there were an estimated 3.8 million patients with a wound managed by the National Health Service (NHS) in 2017/2018, of which 89% and 49% of acute and chronic wounds healed, respectively, within the study year [1]. The impact on the NHS was assessed as £8.3 billion, the majority of this cost incurred in the community, with £5.6 billion spent managing unhealed wounds [1].

The process of wound healing involves four separate, but overlapping stages: haemostasis, inflammation, proliferation and remodelling [2,3,4]. The initial injury triggers vasoconstriction and platelet aggregation [2], releasing cytokines and growth factors which recruit inflammatory leucocytes to the site. Neutrophils enter the wound within less than an hour to eliminate bacteria, followed by macrophages within 48–72 h post-injury, which remove cell and matrix debris and contribute factors which mediate inflammatory responses, enhance angiogenesis and promote the formation of granulation tissue [3,4]. The ensuing formation of a vascular network of capillaries is associated with fibroblast proliferation, collagen deposition and migration of keratinocytes from wound edges, achieving re-epithelialisation [3,4]. Myofibroblasts aid wound contraction, limiting the area for re-epithelialisation, in a remodelling phase which involves degradation of surplus extracellular matrix, replacement of type III with type I collagen, cellular apoptosis and production of new cells [2,3,4].

This natural progression of wound repair is interrupted by increased oxidative stress, unresolved inflammation and immune suppression, resulting in failure to transition into a pro-healing phenotype. Instead, the wound is characterised by hyper-proliferative keratinocytes at the wound edge, and senescent dermal fibroblasts irresponsive to the migratory stimuli [4,5]. Limited numbers of myofibroblasts are present, and fibroblasts and vascular cells undergo apoptosis, leading to loss of granulation tissue [4,5]. Delayed wound healing pathology occurs in immobile and/or aging individuals, associated with co-morbidities, such as diabetes, cardiovascular disease, hypertension, chronic kidney disease and cancer [6,7,8,9], or it can be caused by genetic conditions such as recessive dystrophic epidermolysis bullosa [10]. Clearly, novel therapies, with the ability to effectively promote chronic wound healing in such contexts, represent an unmet global need [1,11].

The first report that showed that high-density lipoproteins (HDL) can help to achieve this therapeutic goal came from Gordts et al. (2014) [12], who demonstrated that topical administration of HDL gel improved delayed (splinted) wound healing in hypercholesterolaemic apolipoprotein (apo)E^−/−^ mice. This finding resonates with reports that exposure to HDL can promote the proliferation of endothelial progenitor cells, aiding wound healing of the arterial wall in hypercholesterolaemic rats [13], and promote wound repair, rescue blood flow and promote neovessel formation due to ischaemia [14,15]; dysfunctional HDL, which arises in a number of disease states, impairs re-endothelialisation in vitro and in vivo [16,17,18,19]. Apolipoprotein (apo) A-I, the major protein component of HDL, and apoA-I mimetics can also promote arterial healing by reducing oxidative stress [20], and protect against impaired re-endothelialisation due to dysfunctional HDL [21], although it cannot entirely replicate the effect of HDL [22]. HDL also stimulates the proliferation and migration of type II alveolar epithelial cells during inflammation [23], and topical administration of synthetic HDL nanoparticles improves corneal re-epithelialisation in diabetic mice following wounding, and in corneas subjected to alkali-burn induced inflammation [24]; these particles also delivered microRNA (miRNA; miR) sequences to epithelial and stromal cells in an intact ocular surface [24].

The ability of HDL to modulate the expression within, and secretion of, selected small non-coding microRNA sequences in cells is well established, achieving regulation of expression of networks of genes, including those involved in lipoprotein metabolism and intercellular communication [25,26,27,28]. Equally, microRNA sequences have been shown to modulate the process of wound healing [29,30,31,32], but insights into the genetic mechanism(s) by which HDL promote this process are lacking are present. In this study, we have investigated the impact of HDL on ‘scratch’ wound healing by keratinocytes, neonatal and adult fibroblasts, validated a microchip array screen of selected microRNA sequences altered in human dermal fibroblasts by HDL treatment, and explored the role of one sequence, hsa-miR-6727-5p, in the process of wound healing in vitro.

## 2. Materials and Methods

### 2.1. Culture and Maintenance of Human Neonatal Keratinocytes and Fibroblasts and Human Adult Fibroblasts

Primary human neonatal keratinocytes (HKn) were obtained from Invitrogen (#C0045C, Paisley, UK), and maintained in serum-free EpiLife™ media (Thermo Fisher Scientific, Waltham, MA, USA) containing S7 supplement (#S-017-5) and penicillin/streptomycin (50 U mL^−1^; 50 μg mL^−1^) and grown on coating matrix (#R-011-K) at 37 °C, 5% CO_2_ from the same supplier. Primary human neonatal dermal fibroblasts (HDFn; #C0045C) were sourced from Invitrogen (Thermo Fisher Scientific, Waltham, MA, USA) and human adult dermal fibroblasts (HDFa) from breast skin samples were provided by the Glasgow Caledonian University Skin Research Tissue Bank (NHS REC Ref 16/E0069); both were maintained in ‘complete’ Dulbecco’s Modified Eagle Medium (DMEM) (Lonza, Basel, Switzerland) supplemented with 10% (*v*/*v*) foetal bovine serum (FBS), L-glutamine (2 mM) and penicillin/streptomycin (50 U mL^−1^; 50 μg mL^−1^) (supplied by Gibco, Thermo Fisher Scientific, Waltham, MA, USA) and incubated at 37 °C, 5% CO_2_. Cells were seeded at a density of 52,000/cm^2^ on 6- and 12-well plates, and 95,000/cm^2^ on 96-well plates, and used between passages 1–4. Cell identity was confirmed by immunofluorescence, using antibodies (AbCAM, Cambridge, UK) against vimentin (1:200 dilution), E-cadherin (1:50) compared with an isotype (IgG) control (1:50) and secondary antibody conjugated to Alex488 (1:750); nuclei were stained using 4′ 6-diamidino-2-phenylindole (DAPI; 10 ng mL^−1^; AbCAM, Cambridge, UK) and images captured using a Zeiss Observer.Z1 AXIO confocal microscope linked to a Zeiss LSM 800 scanner and analysed using the ZEN 2.3 application. Cell viability was assessed using the (3-(4,5-Dimethyl-2-thiazolyl)-2,5-diphenyl-2H-tetrazolium bromide (MTT) assay [33] (Sigma-Aldrich, Gillingham, UK) and an Epoch spectrophotometer (BioTek/Agilent Technologies, Cheshire, UK).

### 2.2. Efflux of [^3^H]Cholesterol to HDL from Human Keratinocytes and Fibroblasts

The human HDL used in this study was supplied by Athens Research and Technology (Athens GA 30601, USA) (https://www.athensresearch.com/products/lipoproteins-and-apolipoproteins/lipoproteins-high-density-human-plasma (accessed on 13 December 2021)); data provided by the supplier indicate a composition of 55–45% lipid, 45–55% protein, and a molecular weight 175,000–360,000, with a purity of ≥95% by electrophoresis. Cholesterol efflux to HDL from primary keratinocytes and fibroblasts was assessed essentially as described in [34,35,36]. In brief, fibroblasts were radiolabelled for 24 h with 1 μCi mL^−1^ [1,2-^3^H (N)]cholesterol (Perkin Elmer, Beaconsfield, UK) in DMEM medium containing 5% (v/v) FBS at 37 °C, 5% CO_2_. The cells were incubated for 18h in DMEM supplemented with 0.1% (*w*/*v*) bovine serum albumin (Sigma-Aldrich, Gillingham, UK) to allow equilibration of [^3^H]cholesterol between differing cellular pools [35,36]. Efflux (24 h) was initiated by the addition of HDL (5–20 μg mL^−1^) in serum-free media, and compared with the basal control; lipids were extracted from cells using hexane:isopropanol (3:2, *v*/*v* (Fisher Scientific, part of Thermo Fisher Scientific; above) and dried at 37 °C under N_2_. The percentage efflux was calculated = (dpm media)/(dpm media + dpm cells) × 100%, as reported previously [34,35,36]. Efflux from keratinocytes was performed in the same way, except that EpiLife media was employed, and the cells were plated onto coating matrix (above).

### 2.3. In Vitro Wound Healing: Scrape (‘Scratch’) Wound Assays

A scrape wound was introduced to confluent (90%) monolayers of fibroblasts or keratinocytes, using a sterile 100 μL pipette tip as previously described [37]. Fibroblasts were incubated in serum-free DMEM, or serum-free media supplemented with HDL (5- 20 μg mL^−1^) or with 10% (*v*/*v*) FBS; keratinocytes were incubated in EpiLife media supplemented with S7 (above) and plated on coating matrix. Cell migration was monitored by capturing triplicate images of wound area (0–72 h) using Image J software (University of Wisconsin) at the intervals indicated in the figure legends, and values normalised by comparison with the corresponding initial wound size.

### 2.4. Total RNA Isolation, Microchip Analysis and Quantitative PCR

Isolation of RNA was performed using either a Directzol RNA MiniPrep Plus kit (Zymo Research, Cambridge Bioscience, Cambridge, UK) or a Nucleospin RNA kit (Machery-Nagel, Germany). Complementary DNA (cDNA) to be used for assessment of gene expression was generated from 250 ng RNA using the High-Capacity cDNA Reverse Transcription kit (ThermoFisher Scientific, Waltham, MA, USA); cDNA for measurement of miRNA and/or mRNA was generated using miScript II RT kit (Qiagen, Manchester, UK) and HiFlex buffer from the same supplier, and performed according to the manufacturer’s instructions. Analysis of miRNA expression was carried out by LC Sciences (Houston, TX, USA): 2632 unique mature hsa-miR sequences derived from miRBase (version 22) [38] were assessed using uParaflo microfluidic chip technology (http://www.lcsciences.com/discovery/applications/transcriptomics/mirna-profiling/mirna/ (accessed on 23 September 2021)). Relative quantitative (q)PCR for miRNA and mRNA expression was performed using HOT FIREPol EvaGreen Q-PCR mix plus (Solid BioDyne), the primer sequences defined in Table 1 and a C1000 thermal cycler with a CFX96 real-time system; no dimer formation was detected using melt curves. Expressions of target miRNA and mRNA were determined using the 2^−^^ΔΔ^^Ct^ method relative to an invariant control sequence or gene, as indicated in the figure legends. Primer efficiencies were confirmed as 90–110% for each sequence investigated. Statistical analysis for significance (*p* < 0.05) used the ΔCt values, compared with the relevant housekeeping sequence.

Delivery of hsa-mir-6727-5p mimic, inhibitor and respective scrambled controls (Qiagen, Manchester, UK) to primary human neonatal and adult fibroblasts was achieved using HiPerFect Transfection Reagent (Qiagen, Manchester, UK), following the manufacturer’s instructions. Transfection complexes were added in serum-free DMEM, and incubated at 37 °C and 5% CO_2_ at the concentrations, and for the periods of time indicated, in the figure legends.

### 2.5. Statistical Analyses

All datasets were deposited in Mendeley Data, V1, doi: 10.17632/mxpjjw4t4g.1. The results are expressed as mean ± SD or SEM of the number of experiments indicated in the figure legends. Statistical analysis was performed by a one-way or two-way ANOVA and Dunnett’s post-test, or a Student’s *t*-test, as indicated in the figure legends; please note that normal distribution was assumed, but could not be proven with the number of independent experiments performed in this study. All testing was performed using GraphPad Prism 8.0 software San Diego, CA, USA; * *p* < 0.05, ** *p* < 0.01 and *** *p* < 0.01.

## 3. Results

### 3.1. The Effect of HDL on Viability of HKn, HDFn and HDFa

Human neonatal keratinocytes showed positive staining for E-cadherin, and negative staining for vimentin, and the isotype control (data not shown); treatment (24 h) with HDL (0–100 μg mL^−1^) had no significant impact on cell viability (Figure 1A), as judged by the conversion of MTT to formazan [33]. Neonatal human dermal fibroblasts stained positively for vimentin and negatively for E-cadherin and the isotype control (data not shown), while exposure to the same range of HDL concentrations revealed significant increases in conversion of MTT to formazan at 5 μg mL^−1^ HDL (21.5 ± 2.43%; *p* < 0.05) and 10 μg mL^−1^ HDL (19.6 ± 6.90%; *p* < 0.05) (Figure 1B) when compared with the control condition. An equivalent staining pattern was observed for HDFa (data not shown); increased conversion of MTT to formazan was observed at 20 μg mL^−1^ HDL (32.3 ± 7.14%; *p* < 0.05) compared to the control (Figure 1C).

### 3.2. Cholesterol Efflux to HDL from HKn, HDFn and HDFa

The responsiveness of human skin cells to HDL was established by measurement of [^3^H]cholesterol efflux to this acceptor, as shown in Figure 2: all cells exhibited dose (5–20 μg mL^−1^) and time dependency of efflux to HDL (Figure 2A,C,E). Human neonatal keratinocytes (HKn) showed a significant (10-1-fold; *p* < 0.01; *n* = 3) increase in efflux of [^3^H]cholesterol to HDL (20 μg mL^−1;^ 24 h) compared with the basal condition (Figure 2B). Under the same conditions, HDL increased efflux from HDFn by 3.82-fold (*p* < 0.01) (Figure 2D), and from HDFa by 6.89-fold (*p* < 0.01) (Figure 2F), compared with the basal condition. No significant differences were noted between basal efflux from HKn, ADFn and ADFa, but efflux from ADFn to HDL (20 μg mL^−1^; 24 h) was significantly lower than from ADFa (*p* < 0.01) or HKn (*p* < 0.05).

### 3.3. The Impact of HDL on Scrape Wound Healing by HKn, HDFn and HDFa

The effects of HDL (5–20 μg mL^−1^) on closure (0–72 h) of scrape wounds in vitro by HKn, HDFn and HDFa are shown in Figure 3. No significant increase in wound closure was noted in the presence of HDL, compared to control, in HKn (Figure 3A), and was, therefore, not investigated further. Exposure of HDFn to HDL enhanced wound closure at 24 h, 48 h and 72 h compared with the control (serum-free media); after 24 h, 10 μg mL^−1^ and 20 μg mL^−1^ HDL increased scrape closure by 1.50-fold (*p* < 0.001) and 1.78-fold (*p* < 0.001), respectively, compared with control (Figure 3B). By contrast, treatment with HDL (5 μg mL^−1^) significantly enhanced wound closure in HDFa (Figure 3C) after 6 h (1.38-fold; *p* < 0.01); this effect was sustained, but not increased, over 72 h (1.32-fold; *p* < 0.001), compared with the control, but no dose dependency was observed. Complete wound closure was achieved in both HDFn and HDFa at 72 h in the presence of media supplemented with 10% (*v*/*v*) FBS (data not shown); analysis of the area under the curve (AUC) revealed no significant differences in wound healing between HDFn and HDFa under basal conditions.

### 3.4. Changes in microRNA Expression Caused by Exposure to HDL in HDFn and HDFa

The changes in expression of microRNA (miRNA) sequences in HDFn and HDFa, caused by exposure (24 h) to 20 μg mL^−1^ HDL compared to the relevant controls (serum-free media) are shown in Figure 4. Treatment with HDL induced pleiotropic changes in miRNA expression, in HDFn and HDFa up-regulating (Figure 4A,D, respectively) and down-regulating (Figure 4B,E, respectively) distinct sequences, compared with the control, visualised using heatmaps [39]; the volcano plots (Figure 4C,F) shows the unstandardised signal (log-fold change) vs. standardised signal (t-statistic) [40], indicating sequences that show alterations (+/−) from the control condition with high significance in HDFn and HDFa, respectively.

### 3.5. Hsa-miR-6727-5p: Regulation by HDL, Delivery and Efficacy

Stimulation of wound healing by HDL in HDFa was less extensive than in HDFn; three sequences exhibiting regulation (hsa-miR-6727-5p, hsa-miR-4787-5p, hsa-miR-3665), and one sequence which did not (hsa-miR-21p), from the miRNA screen were selected for further investigation by qPCR in three independent experiments (Figure 5A,D, respectively). Expression of hsa-miR-6727-5p in response to HDL was significantly increased in HDFn by 33.7% (*p* < 0.05) (Figure 5A) and decreased in HDFa by 26.4% (*p* < 0.05); levels of hsa-miR-3665-5p were also decreased in HDFa by 29.9% (*p* < 0.05) by HDL treatment (Figure 5D). Transient transfection (24 h) of hsa-mir-6727-5p mimic (10 nM) increased the expression of this sequence by 2.2-fold (*p* < 0.01) and 2.87-fold (*p* < 0.01) in HDFn and HDFa, respectively, compared to the scrambled SiRNA control (Figure 5B,E); introduction of the hsa-mir-6727-5p inhibitor (50 nM; 24 h) reduced the expression of this sequence by 44% (*p* < 0.05) and 32.2% (*p* < 0.05), compared with the inhibitor control, in HDFn and HDFa, respectively. The impact of hsa-miR-6727-5p mimic on the expressions of a number of its target genes in HDFn and HDFa are shown in Figure 5C,F: the gene encoding periaxin (*PRX*) was repressed in HDFn (33.9%; *p* < 0.05) and HDFa (36.3%; *p* < 0.01). The hsa-miR-6727-5p mimic also repressed the expression of the genes encoding ER degradation enhancing alpha-mannosidase like protein (*EDEM1*) (49%; *p* < 0.05) and Ki-ras2 Kirsten rat sarcoma viral oncogene homolog (*KRAS*) (56%; *p* < 0.01) in HDFa (Figure 5F).

### 3.6. Hsa-miR-6727-5p Mimic and Inhibitor: Impact on Cell Viability and Scrape Wound Healing in HDFn and HDFa

The impacts of transient delivery of hsa-miR-6727-5p mimic and inhibitor on the viability of HDFn and HDFa, as judged by conversion of MTT to formazan, are shown in Figure 6A,D. Treatment with mimic (10 nM; 24 h) caused a minor, but statistically significant (4.6%; *p* < 0.05) increase in the production of formazan in HDFn, but not HDFa, compared with the scrambled siRNA control; no changes in viability were noted after treatment with the inhibitor sequence (50 nM; 24 h) compared with the same concentration of its control. The effects of treatment with hsa-mir-6727-5p mimic in the scrape wound healing assay in HDFn and HDFa are shown in Figure 6B,E. In the neonatal HDF addition of the mimic maximally increased wound healing after incubation for 9 h (1.47-fold; *p* < 0.05) with stimulation at time points examined thereafter (Figure 6B). By contrast, the mimic significantly reduced wound closure in HDFa, maximally at 9 h (44%; *p* < 0.01), with inhibition sustained after 72 h (Figure 6D). The same inverse relationship was noted in response to hsa-miR-6727-5p inhibitor in HDFn and HDFa (Figure 6C,E): the inhibitor reduced wound healing from 24–72 h by around 25% in HDFn, but significantly (*p* < 0.001) promoted wound closure in HDFa at each time point tested, maximally at 3 h (2.1-fold).

## 4. Discussion

There is an urgent need for the development of new therapeutics to aid chronic wound healing, based on understanding of the mechanisms which promote progression of this process. This study demonstrates the impact of HDL, at concentrations which induce cholesterol efflux (Figure 2), on scrape wound healing in vitro by primary human neonatal keratinocytes, neonatal human dermal fibroblasts and adult human dermal fibroblasts (Figure 3). Improvements in wound closure were noted in cultures of HDFn and HDFa, with the former proving more dose responsive to HDL treatment. Incubation with HDL induced multiple changes in miRNA sequences in both HDFn and HDFa, as determined by microchip array (Figure 4), although some of these changes could not be validated by qPCR (Figure 5). Intriguingly, the expression of hsa-miR-6727-5p was significantly enhanced in HDFn, but decreased in HDFa exposed to HDL; delivery of a hsa-miR-6727-5p mimic elicited differential repression of target genes in HDFn (*ZNF584*) and HDFa (*EDEM3*, *KRAS*). No marked changes in cellular viability were noted in the presence of hsa-miR-6727-5p mimic or inhibitor; notably, promotion of wound closure was observed in the presence of the hsa-miR-6727-5p mimic in HDFn, and by the inhibitor in HDFa (Figure 6).

Effective cholesterol homeostasis in keratinocytes and dermal fibroblasts is needed to sustain the integrity of cutaneous permeability barrier function [41,42,43,44] and to provide one of the key membrane components needed for cellular proliferation and migration [45,46,47,48]. Here, cholesterol efflux was used primarily as confirmation of cellular responsiveness to HDL (Figure 2), but some interesting findings emerge, most notably that efflux from neonatal dermal fibroblasts was lower than that observed in adult dermal fibroblasts and keratinocytes. Differential expression of many genes is found in neonatal and adult fibroblasts [49], including those encoding proteins involved in the organisation and structure of extracellular matrix, cell adhesion, proliferation and migration; our own work has established differential roles for connexin 43 in mediating wound closure in neonatal, juvenile and adult dermal cells [37]. Changes in cholesterol metabolism in dermal fibroblasts are known to occur in genetic conditions such as mitochondrial disorders [50], and inherited disorders of cholesterol metabolism [51,52], but ageing also decreases the cholesterol content within plasma membrane lipid rafts [53], and enhances the cholesteryl ester mass, of human fibroblasts [54], changes which can induce compensatory increases in cholesterol efflux [55].

Treatment with HDL promoted scrape wound closure in HDFn and HDFa, but not keratinocytes. This implies, but does not prove, that the impact of HDL on wound healing [12] is primarily an event mediated by fibroblasts. It is possible that one of the serum-free media components used to sustain keratinocyte growth in culture may mimic the effect of HDL on wound closure, but not cholesterol efflux (above), or that removal of cholesterol via the efflux pathway may limit the source of this material for cell division. Treatment with HDL does not affect keratinocyte viability, but does increase conversion of MTT to formazan in HDFn and HDFa, a finding which implies increased cell metabolism, mitochondrial function, or cell numbers [33]. The concentration of HDL required to increase wound healing in vivo [12] was substantively higher than those used here: 800 μg protein mL^−1^ of HDL in 20% pluronic F-127 gel, every 2 days for 10 days, increased wound healing in apoE^−/−^ mice by nearly 50%, compared with the pluronic gel control. Yu et al. (2017) also utilised high levels of HDL (100 mg mL^−1^ apoA-I concentration) to achieve increases in proliferation and migration of type II alveolar epithelial cells [23], but Zhang et al. (2010) employed equivalent concentrations to those used here to stimulate proliferation of endothelial progenitor cells and promote wound healing [13], via a pathway which involved activation of phosphoinositide -3-kinase (PI3K), protein kinase B (PKB; Akt1) and cyclin D.

An initial screen of the microRNA sequences modified by HDL treatment in HDFn and HDFa revealed multiple sequences which could potentially impact on the complex process of wound closure. However, not all of the regulated sequences identified in this screen could be validated by qPCR in three independent experiments: this discrepancy may be explained in part by signal intensities in the microarray, as lower threshold levels may elicit false positives [56,57,58]. Use of an intercalating dye for detection of poorly expressed miRNA sequences by qPCR may also result in loss of specificity [59,60,61]. Furthermore, the commercial miRNA detection platform uses the miRBase repository for sequence information, which is regularly updated; this could result in non-alignment of the microarray probe sequences and the corresponding miScript primer sequences used here, which were not publicly available.

However, hsa-miR-6727-5p was confirmed to be up-regulated in HDFn, and down-regulated in HDFa, by treatment with HDL (Figure 5); intriguingly, this was not one of the sequences identified as regulated by HDL or apoA-I in human PANC-1 hybrid 1.1B4 pancreatic beta cells under equivalent conditions [28]. Only one published report exists on this sequence, indicating that it promotes the proliferation, migration and invasion of cervical cancer cell lines [60] suggesting a possible role in mediating the impact of HDL in wound healing. This proved to be the case, despite the fact that multiple miRNA sequences are regulated by exposure to HDL: a rise in this sequence in HDFn promoted wound closure, while inhibition of this sequence achieved the same goal in HDFa. The mechanism by which this is achieved remains unknown. Both cell types exhibit repression of *PRX,* the gene encoding periaxin (Figure 6); while periaxin is expressed in fibroblasts [61,62], this protein is more usually associated with myelination in Schwann cells [63], where it interacts with the dystroglycan-dystrophin-related protein -2 complex that links the cytoskeleton to the extracellular matrix. By contrast, KRAS, the most frequently mutated isoform of the *RAS* proto-oncogene, is modestly increased in HDFn, and repressed in HDFa by the hsa-miR-6727-6p mimic. The RAS/mitogen-activated protein kinase (MAP) signalling pathway has been implicated in metabolic reprogramming [64], and cancer-associated fibroblasts are a key element in the tumour microenvironment, influencing cellular proliferation, matrix deposition and remodelling and crosstalk with infiltrating leucocytes [65]. The only other biologically significant finding was the repression of *EDEM3* in HDFa, but not HDFn: EDEM3 is a mannosidase protein involved in ER quality-control, which recognises misfolded proteins targeted for proteasomal degradation [66]. Fibroblasts with a missense variant of EDEM3 exhibit decreased trimming of Man_8_GlcNAc_2_ isomer B to Man7LCNAc2, a decrease in EIF2AK3 (Eukaryotic Translation Initiation Factor 2 alpha kinase; protein kinase R (PKR)-like endoplasmic reticulum kinase (PERK)) expression in response to tunicamycin, and an impaired unfolded protein response [66]. At present, it is merely speculation as to whether any of these changes in gene expression contribute to the differing responses to hsa-mir-6727-5p in HDFn and HDFa: it is an obvious limitation of this study that overexpression and knockdown of these genes has not been performed to confirm a functional role in wound closure, remaining an interesting avenue for further exploration.

## 5. Conclusions

High-density lipoproteins improve wound healing in both neonatal and adult dermal fibroblasts (human), but not keratinocytes, in a manner which does not seem to be related to the ability of this lipoprotein to facilitate cholesterol efflux. Intriguingly, levels of hsa-miR-6727-5p increased in HDFn, but decreased in HDFa, after exposure to HDL. Delivery of a hsa-miR-6727-5p mimic elicited repression of different target genes in HDFn (*ZNF584*) and HDFa (*EDEM3*, *KRAS*), and promoted wound closure in HDFn. By contrast, a hsa-miR-6727-5p inhibitor promoted wound closure in HDFa. We conclude that HDL treatment exerts distinct effects on the expression of hsa-miR-6727-5p in neonatal and adult fibroblasts, and that this sequence plays differential roles in wound healing in these cell types, but cannot replicate the myriad effects of HDL

## Figures and Tables

**Figure 1 membranes-12-00154-f001:**
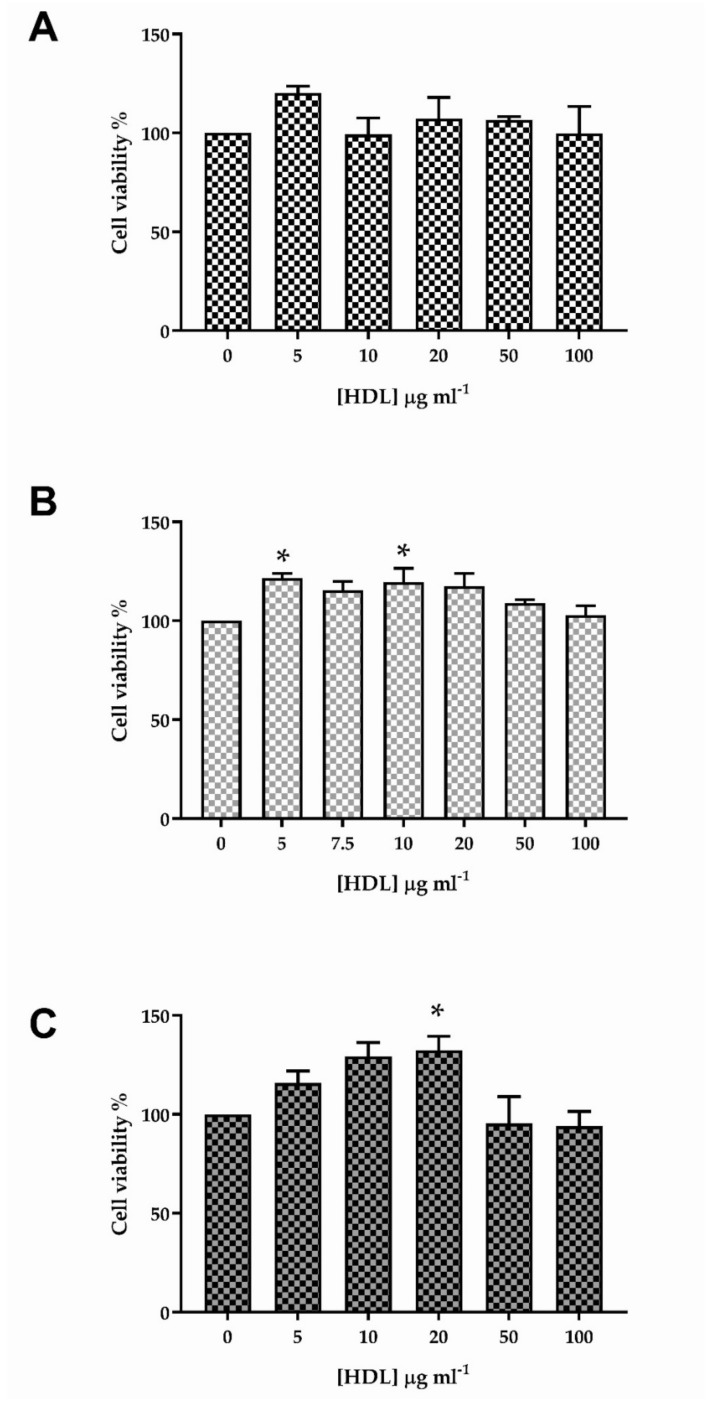
The effects of treatment (24 h) with HDL (0–100 μg mL^−1^) on the conversion of MTT to formazan by HKn, HDFn and HDFa are shown in (**A**–**C**), respectively; data in each figure are the mean ± SEM of three independent experiments, each performed using six replicates. The data were analysed using one-way ANOVA and Dunnett’s post-hoc test; * *p* < 0.05.

**Figure 2 membranes-12-00154-f002:**
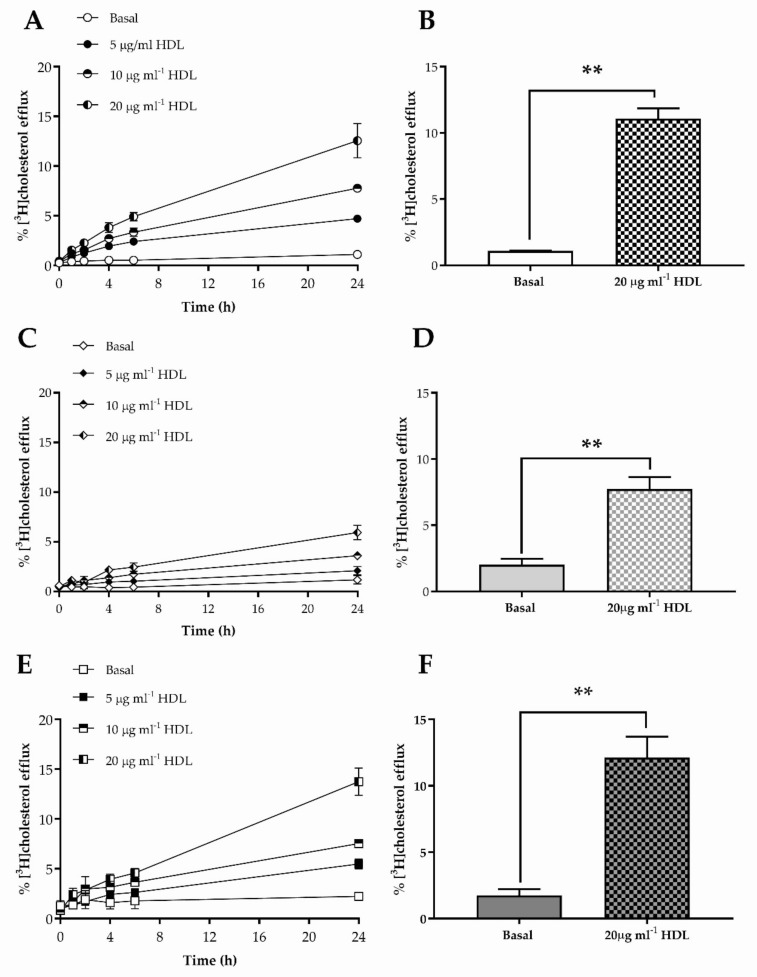
The dose and time dependencies of [^3^H]cholesterol efflux to HDL from HKn, HDFn and HDFa are shown in (**A**,**C**,**E**), respectively: data in each of these figures are from a single experiment and values shown are the mean ± S.D of triplicate wells. (**B**,**D**,**F**) show [^3^H]cholesterol efflux measured in three independent experiments (each performed using triplicate wells), from HKn, HDFn and HDFa, respectively; values are mean ± SEM. The data were analysed using a Student’s *t*-test (paired, two way); ** *p* < 0.01. The data from experiments in (**B**,**D**,**F**) were compared using a one-way ANOVA and Tukey post-hoc test.

**Figure 3 membranes-12-00154-f003:**
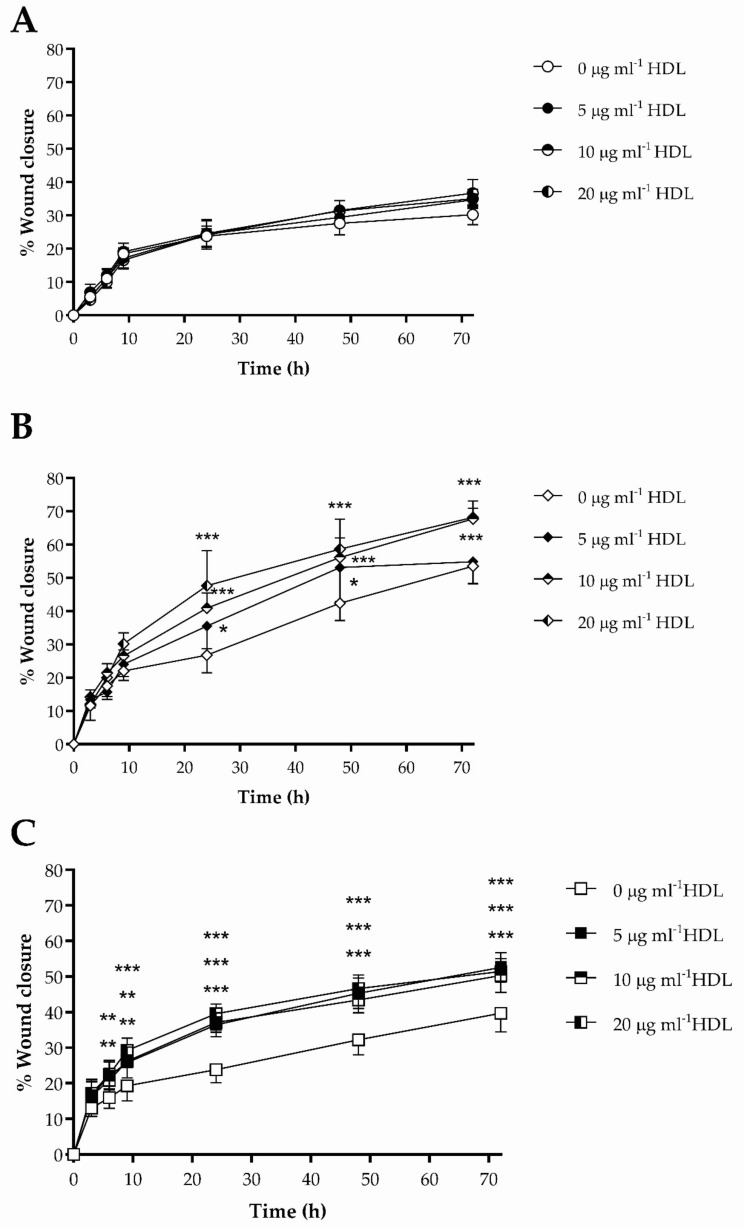
‘Scrape’ wound healing, in the presence or absence of HDL (5–20 μg mL^−1^; 0–72 h), compared with the control (serum-free media) condition, by HKn, HDFn and HDFa, are shown in (**A**–**C**), respectively. Data are the mean ± SEM of three independent experiments (each performed using triplicate wells). The data were analysed using two-way ANOVA and Tukey post-hoc test; * *p* < 0.05; ** *p* < 0.01; *** *p* < 0.001. In (**C**), after 6 h, significance (*p* < 0.01) from control is noted after treatment with HDL (5 μg mL^−1^ and 20 μg mL^−1^); at 9 h, significance is noted for 5 μg mL^−1^ (*p* < 0.01), 10 μg mL^−1^ (*p* < 0.001) and 20 μg mL^−1^ (*p* < 0.001); for time points thereafter, significance (*p* < 0.001) from control is noted for all three concentrations tested.

**Figure 4 membranes-12-00154-f004:**
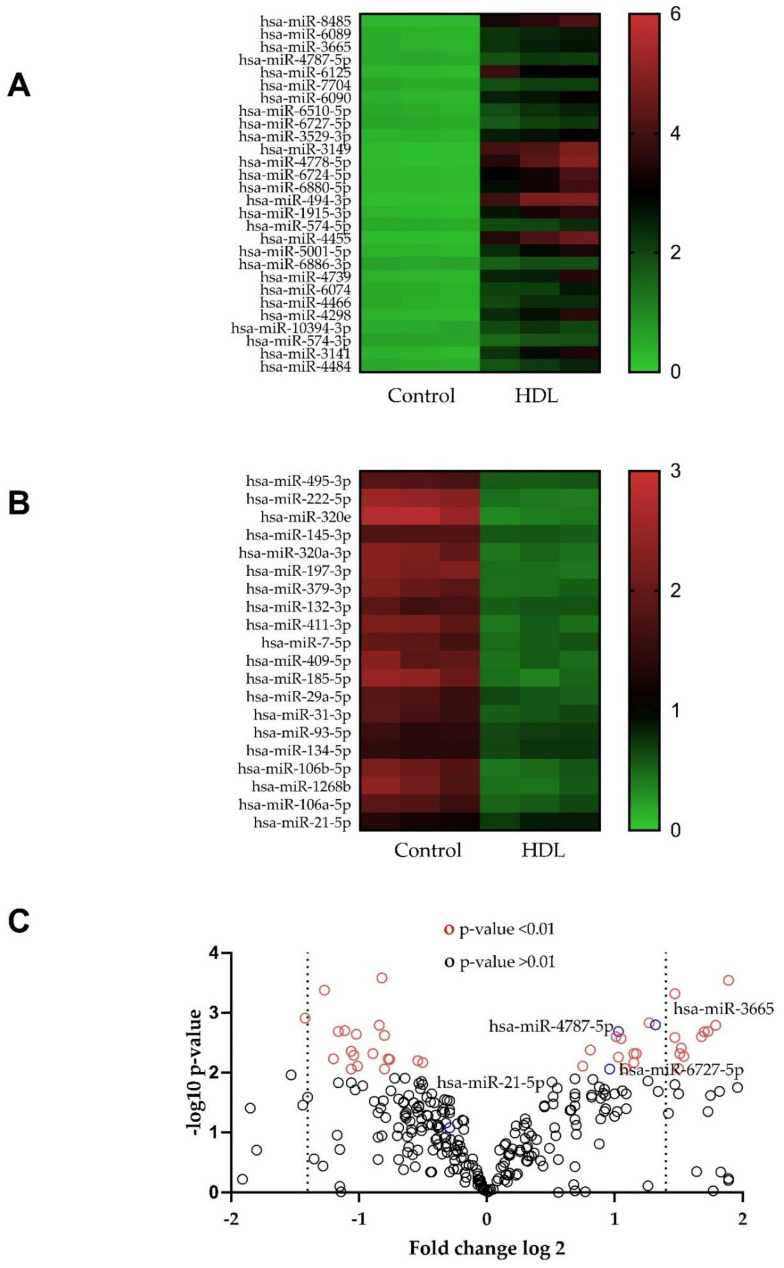
Heatmaps and volcano plots depicting miRNAs regulated following 24 h treatment with 20 μg ml^−1^ HDL, compared with the relevant control, in neonatal human dermal fibroblasts (**A**–**C**) and adult human dermal fibroblasts (**D**–**F**). Data are presented from one sample per group, each with an independent microchip array containing three technical replicates per sequence on each microchip. All miRNAs presented show differences *p* < 0.05 (Student’s *t*-test) compared to the basal control, and are ranked in order of statistical significance.

**Figure 5 membranes-12-00154-f005:**
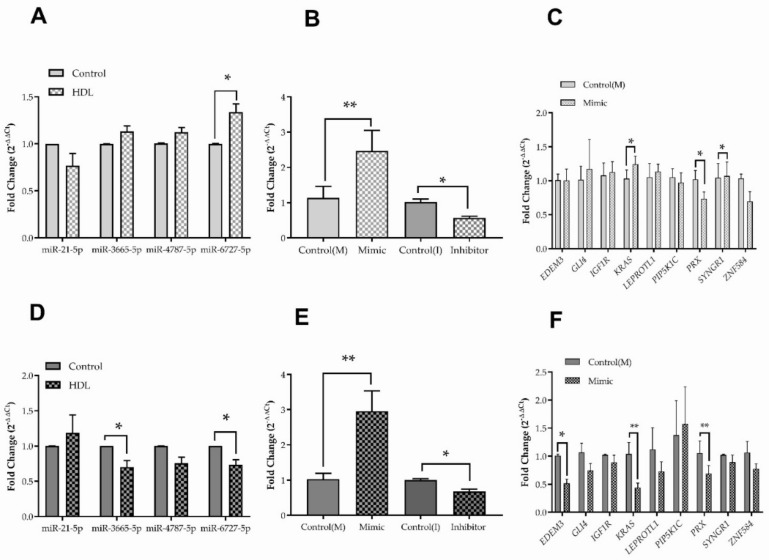
The effect of treatment with HDL (20 μg mL^−1^; 24 h) on fold changes in expression of the miR sequences indicated, in HDFn and HDFa, are shown in (**A**,**D**), respectively, compared with an invariant housekeeping sequence SNORD 72. Values are the mean ± SEM of *n* = 3 (**A**) and *n* = 4 (**C**) independent experiments; statistical analysis was performed using the ΔCt values and a paired Student’s *t*-test. Changes in expression due to delivery (24 h) of 10 nM hsa-miR-6727-5p mimic (M) and 50 nM hsa-miR-6727-5p inhibitor (I) to HDFn and HDFa, compared with the relevant scrambled control sequences, and expressed relative to housekeeping sequence SNORD72 are shown in (**B**,**D**), respectively. Values are the mean ± SEM of *n* = 4 (**B**) and *n* = 3 (**E**) independent experiments; statistical analysis was performed using ΔCt values and a paired Student’s *t*-test. * *p* < 0.05; ** *p* < 0.01. The impact of hsa-miR-6727-5p mimic (10 nM, 24 h), compared with the scrambled control sequence, on the repression of gene expression expressed relative to GAPDH, in HDFn and HDFa are shown in (**C**,**F**), respectively. Values are the mean ± SEM of three independent experiments; statistical analysis was performed using ΔCt values and a paired Student’s *t*-test. * *p* < 0.05.

**Figure 6 membranes-12-00154-f006:**
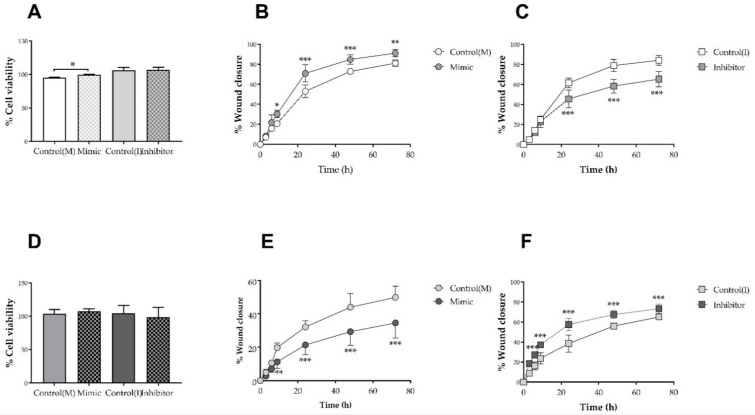
The effects of treatment (24 h) with hsa-mir-6727-6p mimic (10 nM; 24 h) and inhibitor (50 nM; 24 h) on the conversion of MTT to formazan by HDFn and HDFa, compared with the relevant SiRNA scrambled control are shown in (**A**,**D**), respectively; data in each figure are the mean ± SEM of three independent experiments, each performed using six replicates. The data were analysed using a Student’s paired *t*-test; * *p* < 0.05. ‘Scrape’ wound healing (0–72 h) in the presence of the same concentrations of mimic (**B**,**E**) and inhibitor (**C**,**F**) compared with relevant SiRNA scrambled controls, are shown in HDFn (**B**,**C**) and HDFa (**D**,**F**). Data are the mean ± SD of four independent experiments, each performed using duplicate wells; SD was utilised as error bars were too small to indicate the variance of the data using SEM. The data were analysed using two-way ANOVA and Tukey post-hoc test; * *p* < 0.05, ** *p* < 0.01; *** *p* < 0.001.

**Table 1 membranes-12-00154-t001:** Primer sequences employed to measure miRNA sequences and gene expression.

MiRs Primer	Source	miRbase Accession Number	Forward Primer Sequence 5′-3′	Reverse Primer Sequence 5′-3′
miR-1-3p	Human	MIMAT0000416	UGGAAUGUAAAGAAGUAUGUAU	-
miR-6727-5p	Human	MIMAT0027355	CAGGCGGCTGGGA	-
miR-3665	Human	MIMTA0018087	GAGCAGGTGCGGG	-
miR-4787-5p	Human	MIMAT0019956	TGGCGGCGGCA OR GCGGGGGTGGCG	-
miR-21-5p	Human	NR_029493.1	GCAGTAGCTTATCAGACTGATG	-
Gene Primer	Source	NCBI Reference Sequence	Forward Primer Sequence 5′-3′	Reverse Primer Sequence 5′-3′
RNU6	Human	NR-104084.1	TGACACGCAAATTCGTGAAG	-
SNORD-72	Human	NR-002583 (80)	Product code: Hs_SNORD72_11 miScript Primer Assay, Cat. No.: 218300, GeneGlobe ID.: MS00033719	
YWHAZ	Human	NM-001135701.2	ACCGTTACTTGGCTGAGGTTGC	CCCAGTCTGATAGGATGTGTTGG
GAPDH	Human	-	Product code: Hs_GAPDH_1_SG, Cat. no.:249900 GeneGlobe ID.: QT00079247	
RPL13α	Human	NM-012423.4	CTCAAGGTGTTTGACGGCATCC	TACTTCCAGCCAACCTCGTGAG
KRAS	Human	NM-004985	GGACTGGGGAGGGCTTTCT	GCCTGTTTTGTGTCTACTGTTCT
EDEM3	Human	NM-025191	CTCCCTGGCAATCATGCTGAA	AAGCCGGTAAAAGTTTGTAACCT
PIP5K1C	Human	NM-012398	AGACCGTCATGCACAAGGAG	CAGTACAGCCCATAGAACTTGG
ZNF584	Human	NM-173548	GAGGCTCAGTTGGACCCATC	ACATCCCGGTATAGGCCCTTC
IGF1R	Human	NM-000875	ATGCTGACCTCTGTTACCTCT	GGCTTATTCCCCACAATGTAGTT
GLI4	Human	NM-138465	GGAGTCAACTTCGGTCGGAG	TCAGGAGCAGCGAGTTATACT
LEPROTL1	Human	NM-015344	TTTGATGCTTGGATGTGCCCT	GCCCGTTGTAAGAAAGATGGC
SYNGR1	Human	NM-004711	GTGTTCGGCTCCATCGTGAA	GTTGGGGTTGCGGTTGTAGAT
PRX	Human	NM-181882	GGTGGAAATTATCGTGGAGACG	GCAGCTCCCGAACGAAGAT
Universal Reverse Primer	Human	-	-	GAATCGAGCACCAGTTACGC

## Data Availability

The data presented in this study are openly available in Mendeley Data: 10.17632/mxpjjw4t4g.1 (accessed on 29 November 2021).

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
