# Peer review of "Homo Sapiens (Hsa)-microRNA (miR)-6727-5p Contributes to the Impact of High-Density Lipoproteins on Fibroblast Wound Healing In Vitro"

_membranes, 2022, doi:10.3390/membranes12020154_

Round 1

Reviewer 1 Report

The paper describes new details of biochemistry in wound healing. The results appear to be also of some practical significance. The presentation needs some corrections.

The abbreviation in the title should be changed for a definition of this abbreviation (hsa-miR-6727-5p). The title should give an idea about the content also for the non-specialist Reader.

It appears to the Referee that it would be better to put the List of  Abbreaviations after the Keywords.

The Abstract and the Conclusions appear to the present Referee contradictory. I seems not sufficiently clear what influences what?

The Reference list should be checked again to avoid typing (eventual other?) errors like in Ref. 4. by writing "cellular" instead of Cellular.

Reviewer 2 Report

  1. in Martials and Methods countries of manufacturers’ origin are missing
  2. several manufacturers’ names and country of origin are missing e.g. media and supplements, antibodies
  3. The authors did not provide name and manufacturer of instrument, which they use in MTT assay
  4. Lines 86-89 are highlighted.
  5. The authors use parametric statistical tests. Did they confirmed, that distributions of variables were parametric? If yes, please include this in descriptions of statistical analysis, if not please implement.
  6. The quality of figures pasted into manuscript is relatively very low, please provide high resolution figures
  7. Table 1 is mentioned, but is missing
